# Measurement in the study of menstrual health and hygiene: A systematic review and audit

**Julie Hennegan**[1]*, **Deborah Jordan Brooks**[2], **Kellogg J. Schwab**[1], **G. J. Melendez-Torres**[3]

**1** The Water Institute, Department of Environmental Health and Engineering, Johns Hopkins Bloomberg School of Public Health, Baltimore, Maryland, United States of America, **2** The John Sloan Dickey Center for International Understanding and Department of Government, Dartmouth College, Hanover, New Hampshire, United States of America, **3** Peninsula Technology Assessment Group, College of Medicine and Health, University of Exeter, Exeter, United Kingdom

* jhenneg1@jhu.edu

## Abstract

**Data Availability Statement:** All relevant data are within the paper and its Supporting Information files.

### Background

The lack of established measurement tools in the study of menstrual health and hygiene has been a significant limitation of quantitative studies to date. However, there has been limited exploration of existing measurement to identify avenues for improvement.

### Methods

We undertook two linked systematic reviews of (1) trials of menstrual health interventions and their nested studies in low- and middle-income countries, (2) studies developing or validating measures of menstrual experiences from any location. Systematic searching was undertaken in 12 databases, together with handsearching. We iteratively grouped and audited concepts measured across included studies and extracted and compared measures of each concept.

### Results

A total of 23 trials, 9 nested studies and 22 measure development studies were included. Trials measured a range of outcomes including menstrual knowledge, attitudes, and practices, school absenteeism, and health. Most measure validation studies focused on assessing attitudes towards menstruation, while a group of five studies assessed the accuracy of women's recall of their menstrual characteristics such as timing and cycle length. Measures of menstrual knowledge, attitudes, beliefs and restrictions were inconsistent and frequently overlapped. No two studies measured the same menstrual or hygiene practices, with 44 different practices assessed. This audit provides a summary of current measures and extant efforts to pilot or test their performance.

**Funding:** This study was funded by The Case for Her (http://www.thecaseforher.com/) and the Osprey Foundation of Maryland (http://www.ospreyfdn.org/) (JH, KS). The corresponding author (JH) had full access to all the data in the study and had final responsibility for the decision to submit for publication. The funders of the study had no role in study design, data collection and analysis, decision to publish, or preparation of the manuscript.

**Competing interests:** The authors have declared that no competing interests exist.

## Conclusions

Inconsistencies in both the definition and operationalisation of concepts measured in menstrual health and hygiene research should be addressed. To improve measurement, authors should clearly define the constructs they aim to measure and outline how these were operationalised for measurement. Results of this audit indicate the need for the development and validation of new measures, and the evaluation of the performance of existing measures across contexts. In particular, the definition and measurement of menstrual practices, knowledge, attitudes, norms and restrictions should be addressed.

## Review protocol registration

CRD42018089884.

## Introduction

Menstruation is a recurring experience in the lives of millions of women and adolescent girls across the globe. This natural process has gained increased attention for its role in female health and social participation in recent years, following a history of neglect and silence. Policy and programming have rapidly expanded, seeking to address unmet menstrual needs. This response has far outstripped the pace and funding of research to understand menstrual experiences and inform and evaluate interventions.[1] A growing body of qualitative research has highlighted the challenges faced by menstruating women and adolescent girls in low-resource settings and indicated negative effects on health, education, employment, and well-being. These studies have also highlighted a complex array of factors contributing to experiences of menstruation.[2] Quantitative studies testing risk factors or consequences hypothesized through qualitative research are rare, and systematic reviews of quantitative and trial research have highlighted the limited number and low quality of extant studies.[3–5]

A significant challenge for quantitative research of menstrual health has been a lack of clarity around core concepts and a paucity of measurement tools to capture them. In 2016, Hennegan and Montgomery [3] highlighted this inconsistency in the measurement of outcomes across trials, as well as the absence of relevant menstrual health measures to capture experiences as a key barrier to improved trials of interventions. Similarly, in mapping the knowledge of menstrual health and hygiene across study designs, Chandra-Mouli and Patel [6] noted that *"vague measures are often used to describe the menstrual experiences of girls, which impede data aggregation and direct comparisons."* In proposing research agendas for menstrual health, many have called for improved clarity in the measures used, refinement of concept definitions, consistency in outcomes assessed, and the development and validation of new measures where needed [7–9]. While past reviews and research priority papers have highlighted measurement challenges as barriers, none have reviewed existing measures to provide an appraisal of current use and gaps.

The present study seeks to inform improved measurement in the rapidly emerging field of menstrual health by auditing extant measurement. We aimed to describe the current concepts assessed in studies of interventions and identify the measures that have been used to capture core concepts. Two linked, simultaneous systematic reviews were undertaken. Review A included trials of menstrual health interventions in low- and middle-income countries (LMICs), as well as studies nested within those trials. The aim of the first review was to identify

the concepts and outcomes measured in studies of interventions, supplemented by nested studies which may provide further explorations of menstrual experiences, measures, intervention theory of change or process evaluation. Review B collated and appraised studies developing, validating or testing tools to measure menstrual experiences across geographies. This second review aimed to identify the concepts for which measures have been developed and to highlight lessons learned from measure development and validation. By integrating the two reviews we were able to compare the measures used in trials to those developed and tested to date.

Together, this work provides an overview and appraisal of current measurement in menstrual health and hygiene research and develops recommendations for pathways forward.

## Methods

The review protocol is registered on PROSPERO: [CRD42018089884] and is reported according to PRISMA guidance (S1 Table).

### Search strategy and selection criteria

Two systematic searches were undertaken in English, reported in Box 1. Trial searches were conducted in the following 12 databases: Cochrane Central Register of Controlled Trials (CENTRAL), Cumulative Index of Nursing and Allied Health Literature (CINAHL), ProQuest Dissertation and theses, Embase, Global Health, Medline, Open Grey, Popline, PsycINFO, Social Sciences Full Text, Socoiological Abstracts, WHO Global Health Library. In addition, we screened the first 20 pages of Google Scholar results, and trial registries: Clinical Trials Registry, Pan African Trials Registry, Trials Register of Promoting Health Interventions (TRo-PHI). To identify nested studies, we undertook vertical searching of reference lists and citations of included trials. Searches for measure development and validation studies were undertaken in the same databases but excluded CENTRAL and trial registries.

We searched the reference lists and citations of two past systematic reviews [3, 4] as well as a report reviewing the state of menstrual health research [10]. To identify grey literature, we searched online databases specific to menstrual health, hygiene and sanitation: Menstrual Health Hub, Menstrual Hygiene Day 'Resources', and Sustainable Sanitation Alliance (SuSanA library). Further we searched the websites of key organisations undertaking work in menstrual health: Oxfam, PATH, Plan International, Save the Children, UNICEF, UNFPA, UN Women, WaterAid, WASHUnited, WSSCC.

Initial searches were undertaken in English in March 2018 and updated in July 2019. Titles and abstracts were independently screened by two reviewers using EPPI-reviewer 4, with full text screening undertaken by the first author.

**Review A: Trials audit.** Inclusion criteria for trial design were consistent with those applied in a past systematic review of menstrual health and hygiene intervention studies [3]. Randomized and non-randomized trials which included a control group (including controlled before-after studies) were eligible for inclusion [11]. Trials were eligible if they evaluated the effectiveness of interventions designed to improve the menstrual experiences of women or girls. Interventions could include interventions such as puberty education or social programs designed to improve social support or reduce menstrual stigma. Interventions providing supportive resources and environments were also eligible, such as the provision of menstrual products (e.g., sanitary pads) or improvements to water, sanitation and hygiene (WASH) infrastructure. We included studies that compared the acceptability, comfort or experience of using different menstrual products when they met study design criteria. Nested studies were eligible if they included quantitative data on the menstrual experiences or environments of

## Box 1. EMBASE search strategies

| Search A: Trial audit | |
|---|---|
| **Search 1:**<br><br>Menstrual | 'menstruation'/exp OR 'menarche'/exp OR<br><br>('menstrual period' OR menstru* OR menses OR catamenia OR menarche):ti,ab,kw |
| **Search 2:**<br><br>Interventions | (intervention* OR hygiene OR hygienic OR education OR information OR 'software' OR 'hardware' OR management OR product* OR absorb* OR material* OR WASH OR water OR sanitation OR sanitary OR toilet* OR latrine* OR privy OR water closet OR lavatory OR 'girl friendly' OR 'wom?n friendly' OR 'gender separate' OR 'gender-separate' OR privacy OR private OR dispos* OR waste):ti,ab,kw |
| **Search 3:**<br><br>Trial method | 'intervention study'/exp OR 'community trial'/exp OR 'controlled clinical trial (topic)'/exp OR<br><br>('randomized controlled trial' OR 'randomised controlled trial' OR 'controlled trial' OR 'control trial' OR 'controlled study' OR 'control group' OR 'clinical trial' OR 'experimental study' OR 'non-randomized trial' OR 'non-randomised trial' OR 'pilot trial' OR randomi?ed OR randomi?e OR trial OR randomly OR evaluation):ti,ab,kw |
| **Final search:** | 1 AND 2 AND 3 |
| Search B: Measure studies | |
| **Search 1:**<br><br>Menstrual | 'menstruation'/exp OR 'menarche'/exp OR<br><br>('menstrual period' OR menstru* OR menses OR catamenia OR menarche):ti,ab,kw |
| **Search 2:**<br><br>Behaviours and experiences | 'social behavior'/exp OR 'experience'/exp OR 'comprehension'/exp OR 'satisfaction'/exp OR<br><br>(experience* OR practice* OR behaviour* OR behavior* OR management OR 'menstrual hygiene' OR 'menstrual health' OR attitude* OR perspective* OR perception* OR preference OR satisfaction OR views):ti,ab,kw |
| **Search 3:**<br><br>Measure development | 'measurement repeatability'/exp OR 'measurement precision'/exp OR 'measurement accuracy'/exp OR 'validation study'/exp OR<br><br>(measure OR 'measurement tool' OR scale OR indicat* OR assessment OR tool OR questionnaire OR survey OR psychometric* OR validat*):ti,kw |
| **Final Search:** | 1 AND 2 and 3 |

participants. Nested qualitative studies were not eligible. Studies of menstrual experiences nested in trials which did not meet inclusion criteria, or trials not yet reported (e.g., only baseline data available) were not included. Studies including women and girls of reproductive or pre-reproductive age in low- and middle-income countries were eligible [12]. We were unable to include studies that were not available in English or Spanish.

**Review B: Measure studies.**   Studies were eligible if they reported on the development and validation, or tested the performance of, measures of menstrual experiences. The aim of this review was to inform future measurement in the study of menstrual health and hygiene in low-resource settings and we sought to identify any measures of menstrual management behaviours, attitudes, knowledge or the impact of menstruation on quality of life. To align our review of measures with menstrual health and hygiene research and interventions focused on

experiences of non-disordered menstruation, we excluded measures designed to diagnose menstrual disorders such as endometriosis or heavy menstrual bleeding. Measures that focused on symptoms associated with hormonal cycling, that is, physical symptoms associated with menstruation, were also excluded. Similarly, we excluded measures of the experiences of disorders of the menstrual cycle including: Polycystic Ovary Syndrome (PCOS), dysmenorrhea, or endometriosis. We excluded measures of the acceptability of menstrual suppression or experiences of menopause. Studies from any country were eligible.

## Analysis and quality appraisal

Review A and Review B study data were extracted using piloted forms by one reviewer and checked by a second. For Review A we extracted concepts measured as described by each study's authors and iteratively grouped the concepts measured. Within each grouping we summarized the ways different concepts had been described and measured. Where described we extracted measure specifics, such as piloting or validation, question wording or recall periods, to further inform comparisons. As the aim of this review was to audit the measures used in trial, we did not assess risk of bias in trial designs.

Through Review B we collated the measures developed to date and the contexts in which these had been assessed. Included measure studies ranged from those assessing the accuracy of single self-report questions, to those developing scales relating to latent constructs. Where latent constructs had been described, we extracted authors' definitions and any subscales along with example items from the measures. We grouped Review B measures according to the iterative conceptual groupings developed in Review A. This enabled us to describe the availability of measures for the concepts of focus in intervention research. We did not undertake formal quality appraisal of measure development studies, as the concepts measured were poorly aligned with trial outcomes, requiring more attention to concept analysis. We supplemented this with brief summary of the performance of the developed measures and the concepts studies used to establish convergent, discriminant or predictive validity as this is likely to be most useful for future measure development efforts.

## Results

Searching and screening results are presented in Fig 1.

## Included studies

Characteristics of included studies are summarized in Tables 1 and 2.

Among the trial studies, 13 tested effects of education interventions, 8 tested product provision interventions or compared comfort and acceptability of different products (n = 4), and two tested combined education and product provision. Products tested included disposable and reusable pads (n = 6), menstrual cups (n = 2) and a combination of pads and cups (n = 2). Most studies focused on girls in schools or of school age (n = 18), with an additional 2 studies undertaken with university students. One study exclusively concerned girls who reported experiencing menstrual pain. One study sought to provide training for adolescents with mild intellectual disability (IQ 50–70) and their parents. No other studies reported including participants with disabilities. Nested studies explored intervention effects and menstrual experiences in more detail, with study aims reported in Table 1.

Of measure development studies, 11 developed a scale or item set to measure a defined latent construct, 6 investigated the performance, revalidation or a new language version of an existing measure, and 5 evaluated the accuracy of self-reported menstrual characteristics such as the date of menarche. Six studies were published before 2000, nine in the 2000's and a

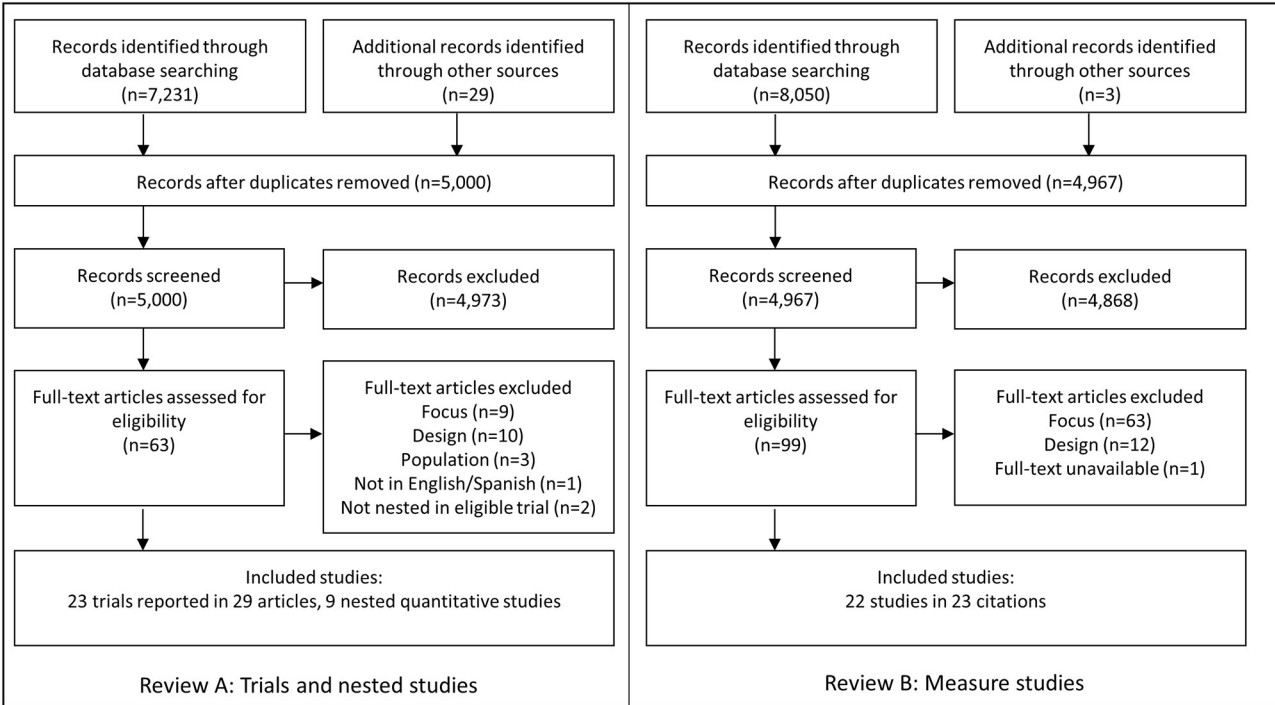

**Fig 1. Flow diagrams reporting titles, abstracts and full texts screened for Review A and Review B searches, along with reasons for exclusion of full text articles.**

further seven since 2016. More studies were conducted in North America than in other regions (n = 10), followed by Europe (n = 4). There were two studies each with samples from Iran and India, and single studies undertaken in Turkey, El-Salvador and the Philippines, Belize, Mexico and Israel. Many studies were undertaken with girls in schools (n = 9), and university students (n = 8).

## Concepts measured across included studies

From Review A we included measures used as outcomes as well as those used descriptively to contextualise participant experiences and in nested studies. Concepts described by each study's authors, their methods of assessment and a brief summary of measure development or validation efforts are reported in S2 Table. Most measures were designed by investigators for the purpose of their individual study, with some referring to external experts or past research to inform questions. Eight trials reported piloting measures prior to data collection or were themselves pilots to assess the feasibility of the measures tested. To describe and compare measures we iteratively grouped concepts measured across studies, displayed in Table 3.

From Review B we extracted the measures developed, their sub-scales and example items from each, reported in Table 4. To compare developed measures from Review B with measures in trials (Review A) we highlight in Table 4 where measures, or sub-scales within measures, assess concepts corresponding to those in trials and nested studies. For studies developing or testing scales we extracted reported tests of dimensionality, reliability and validity and report these in S3 Table. Across scale development and validation studies most investigated measure factor structure, although many employed only exploratory factor analysis and did not include a confirmatory analysis. Internal consistency was reported for most measures as the sole test of

**Table 1. Review A included study characteristics.**

| Study ID | Data collection dates | Country, region (Urban/Rural) | Sample size (clusters) | Sample characteristics | Study design | Study aim |
|---|---|---|---|---|---|---|
| Abedian 2011 [13] | Apr–Oct 2009 | Iran, Mashhad (Urban) | 165 | Age 19–25 experiencing dysmenorrhea and living in university dormitories | Individual-randomized controlled trial | Compare impact of peer-led and health-provider-led self-care education on girls' knowledge, attitudes and dysmenorrhea symptoms. |
| Beksinska 2015 [14] | Jan–Nov 2013 | South Africa, Durban (Urban) | 110 | Age 18–45 using contraception but with regular menstrual cycles, had water and no STI. Recruited from reproductive health clinic | Cluster randomized controlled trial | Evaluate acceptability and performance of a menstrual cup compared to tampon or pad use. |
| Blake 2018 [15] | NR | Ethiopia, West Shewa, (Rural) | 636 (20 schools) | Age 10–19 in schools | Cluster randomized controlled trial | Assess the impact of the Ethiopia Growth and Changes puberty book on girls' menstrual hygiene at the individual, community and environment levels. |
| Dhanalakshmi 2015 [16] | NR | India, Tamil Nadu, Vellore (Urban) | 62 | Age 10–19 inpatient or outpatients at tertiary care centre | Individual-randomized controlled trial | Test adolescent awareness of reproductive health and evaluate effectiveness of structured teaching program on knowledge and beliefs regarding menstruation, pregnancy and sexual behaviour. |
| Djalalinia 2012 [17] | NR | Iran, Tehran (Urban) | 1823 (15 schools) | Aged 11–15 middle school students | Individual-randomized controlled trial | Test effects of a health education intervention delivered by either trainers or parents on menstrual health promotion (menstrual experience and hygiene practices). |
| El-Mowafy 2014 [18] | Oct 2012–Mar 2013 | Egypt, Damietta City (Urban) | 234 (7 orphanages) | Aged 14–16 girls in orphanage homes | Individual-randomized controlled trial | Test effects health education on menstrual knowledge and practices. |
| Fakhri 2012 [19] | NR | Iran, Mazandaran province (Urban & Rural) | 689 | Aged 14–18 high school girls in schools with low socio-economic status | Controlled before-after study | Test effects of an educational program on the health and hygiene of girls during menstruation. |
| Fetohy 2007 [20] | NR | Saudi Arabia, Riyadh (Urban) | 248 | Secondary school grades 1–2 (majority 14–16) | Cluster randomized controlled trial | Assess impact and suitability of a menstrual education program on menstrual knowledge, attitudes and practices. |
| Leventhal 2016 [21–23] | NR | India, Bihar (Rural) | 3,363 (76 schools) | Mean age 12.97 years, middle school girls in 7th and 8th grade | Cluster randomized controlled trial | Factorial trial to test the impact of resilience curriculum, health curriculum, and combined curriculum compared to usual curriculum on health, emotional and well-being outcomes. |
| Mbizvo 1997 [24, 25] | NR | Zimbabwe (Urban & Rural) | 1689 (856 female) | Aged 10–19+ (Mean = 14.6), secondary school students | Cluster randomized controlled trial | Assess impact of school health education on reproductive health knowledge, menstrual knowledge, attitudes and practices. |
| Mohamed 2014 [26] | NR | Malaysia, Penang (Urban) | 18 (repeated for each condition) | Unmarried Muslim (Mean age 25) females recruited from research institute with no reported menstrual pain. | Individual-randomized cross-over trial | Measure physiological and psychological responses to sanitary pads of different thickness across activity levels. |
| Montgomery 2012 [27, 28] | 2008–2009 | Ghana, Central & Ashanti regions (Urban and Rural) | 120 (4 schools) | Aged 12–18 at joined primary and junior secondary schools | Non-randomised cluster-controlled trial | Investigate the relationship between the availability of sanitary-pads and education, and education alone on school attendance among girls. |

(*Continued*)

**Table 1.** (Continued)

| Study ID | Data collection dates | Country, region (Urban/Rural) | Sample size (clusters) | Sample characteristics | Study design | Study aim |
|---|---|---|---|---|---|---|
| Montgomery 2016 [29] | Jan 2012–Dec 2014 | Uganda, Kamuli district (Rural) | 1124 (8 schools) | Grades 3–5 in primary schools (10–13+ at baseline) | Non(quasi)-randomised cluster-controlled trial | Assess impact of providing reusable sanitary pads and puberty education on girls' school attendance and psychosocial well-being. |
| Hennegan 2016a [30] | Nov 2014 | Uganda, Kamuli district (Rural) | 205 (8 schools) | Aged 10–19, menstruating girls in primary schools. | Nested cross-sectional survey using trial endline data | Describe girls' experiences of the reliability and acceptability of different menstrual materials and self-reported freedom of activity according to the material used. |
| Hennegan 2016b [31] | Nov 2014 | Uganda, Kamuli district (Rural) | 205 (8 schools) | Aged 10–19, menstruating girls in primary schools. | Nested cross-sectional survey using trial endline data | Describe girls' menstrual hygiene management (MHM) practices and contribution of MHM to health, education and psychosocial experiences. |
| Oster 2011 [32] | Nov2006–Jan 2008 | Nepal, Chitwan district (Rural) | 198 (4 schools) | Grades 7–8 (Mean age 14.2) | Individual-randomized controlled trial | Evaluate effects of provision of menstrual cups on girls' school attendance. |
| Oster 2012 [33] | Nov2006–Jan 2008 | Nepal, Chitwan district (Rural) | 198 (4 schools) | Grades 7–8 (Mean age 14.2) | Nested longitudinal study + secondary analysis of trial data | Estimate the role of peer effects in the adoption of the menstrual cup during randomized controlled trial. |
| Phillips-Howard 2016 [34] | Aug2012–Nov 2013 | Kenya, Gem District (Rural) | 751 (30 schools) | Age 14–16 menstruating primary school girls | Cluster randomized controlled trial | Test the effect of menstrual hygiene on schoolgirls' school and sexual/reproductive health outcomes. Evaluate impact of providing menstrual cups, sanitary pads or control. |
| Nyothach 2015 [35] | Aug2012–Nov 2013 | Kenya, Gem District (Rural) | 723 (30 schools) | Age 14–16 menstruating primary school girls | Nested comparison across cluster randomized controlled trial groups | Compare Water, Sanitation and Hygiene (WASH) observations and self-reported handwashing behaviour across groups using different menstrual materials. |
| Oduor 2015 [36] | Aug2012–Nov 2013 | Kenya, Gem District (Rural) | 766 (30 schools) | Age 14–16 menstruating primary school girls | Nested longitudinal study | Describe changing and disposal practices, particularly dropping materials, across menstrual materials used during the trial. |
| Juma 2017 [37] | Aug2012–Nov 2013 | Kenya, Gem District (Rural) | 604 (30 schools) | Age 14–16 menstruating primary school girls | Nested cross-sectional study at endline | Explore menstrual cup safety, testing emergence of adverse health outcomes and Staphylococcus aureus vaginal colonization or Escherichia coli growth on sampled used cups. |
| van Eijk 2018 [38] | Aug2012–Nov 2013 | Kenya, Gem District (Rural) | 192 (10 schools) | Age 14–16 menstruating primary school girls | Nested longitudinal study of cup condition | Compare self-reported cup use to observed cup colour change, and examine factors influencing time to uptake. |
| Alexander 2018 [39] | Jun 2012–Oct 2013 | Kenya, Gem District (Rural) | 30 schools | Rural primary schools | Nested repeat cross-sectional survey | Compare school WASH conditions (observational checklist) between baseline and follow-up across intervention conditions. |
| Benshaul-Tolonen 2019 [40] | Oct 2012–Nov 2013 | Kenya, Gem District (Rural) | 751 (30 schools) | Age 14–16 menstruating primary school girls. Spot-check and register data from full school sample. | Cluster randomized controlled trial findings and nested study | Test the effect of menstrual cup and sanitary pad provision interventions to control on school attendance. |

(*Continued*)

**Table 1.** (Continued)

| Study ID | Data collection dates | Country, region (Urban/Rural) | Sample size (clusters) | Sample characteristics | Study design | Study aim |
|---|---|---|---|---|---|---|
| Sener 2019 [41] | Apr 2015–Feb 2016 | Turkey, Duzce (Urban) | 28 girls (1 school) | Girls with mild intellectual disability (Mean age 17.4) | Individual-randomized controlled trial | Testing personal hygiene training program for teenagers with mild intellectual disability and their parents on knowledge, skills, attitudes and behaviour. |
| Setyowati 2019 [42] | May 2018 | Indonesia, Aceh Besar district | 174 | Aged 9–12 pre-menarche girls | Controlled before-after study | Identify influence of reproductive health education on females' preparation, knowledge, emotional response and attitude towards menarche. |
| Sharma 2015 [43] | NR | India, Dehradun | 50 | Aged 11–17 menstruating high school students | Controlled before-after study | To assess knowledge and practice regarding menstrual hygiene before and after a teaching program. |
| Shrestha 2018 [44] | NR | Nepal, Nuwakot district | 716 | Aged 15–50 | Individual-randomized controlled trial | Compare roles of awareness and cost on demand for sanitary pads. |
| Stadler 2006 [45] | NR | Nigeria, Abuja (Urban) | 289 | Aged 18–45 currently using commercial pads, no menstrual disorders. | Individual-randomized controlled trial | Compare the acceptability, comfort and reliability of top-sheet compared to non-woven sanitary pads. |
| Valizadeh 2017 [46, 47] | 2014 | Iran, Tabriz (Urban) | 364 (12 schools) | Aged 11–14 menstruating girls in secondary schools | Cluster randomized controlled trial | Compare effects of educating mothers and daughters, daughters alone, or no education on knowledge, attitudes and practices of puberty hygiene. |
| Wilson 2014 [48, 49] | NR | Kenya, Nyaza province (Rural) | 302 (10 schools) | Mean age 15.5 girls in primary and secondary schools | Partial preference cluster-randomized controlled trial | Assess impact of menstruation on school attendance and evaluate acceptability of training girls to construct reusable menstrual products. |
| WoMena 2018 [50] | Jan–Jun 2017 | Uganda, Rhino Camp (Rural) | 55 | School girls (Mean age 16), and adult guardians in refugee settlement | Controlled before-after study | Assess the acceptability and feasibility and potential health and social impact of introducing menstrual cups and reusable pads in a refugee settlement context. |

NR: Not Reported; WASH: Water, Sanitation and Hygiene; SD: Standard Deviation.

reliability. Of the 11 studies testing new measures, only three reported test-retest reliability. Most studies appraised face and content validity through expert and participant input, or comparison to qualitative study findings. Far fewer studies included further quantitative validation such as tests of convergent, discriminant or predictive validity. This evidenced the lack of a clear nomological network, theory about the interrelationship between constructs and their measures, to guide the selection of related constructs. One measure of attitudes assessed relationships between the scale and mental health, self-esteem and locus of control [71]. While a second tested relationships with self-objectification [68]. A final instrument testing a broad menstrual health construct, assessed criterion validity against self-reported menstrual symptoms and quality of life measures [69].

## Menstrual and hygiene practices

**Review A.** Of 32 included trials and nested studies, 19 reported measuring different practices undertaken to manage menses, not all of these were considered 'hygiene' behaviours and so are referred to here and menstrual and hygiene practices. In studies, these were typically

**Table 2. Review B included study characteristics.**

| Study ID | Country, region (Urban/Rural) | Sample size (clusters) | Sample characteristics | Measure tested | Study aim |
|---|---|---|---|---|---|
| Alimoradi 2019 [51] | Iran (Urban) | 560 (52 high schools) | School girls ages 13–19 | Female adolescents' sexual reproductive self-care scale | Design and evaluate the psychometric properties of an instrument for understanding female adolescents' reproductive and sexual self-care behaviours. |
| Aubeeluck 2002 [52] | U.K, Bedfordshire (Urban) | 100 | Convenience sample of university students and staff. Mean age = 28.2. | Order/priming effects. Menstrual Attitudes Questionnaire (MAQ) | Test the effect of priming using a 'menstrual joy questionnaire' and order of administration on MAQ scores. |
| Bargiota 2016 [53] | Greece, NR | 301 | Women aged 18–45, excluding pregnant women | MAQ | Test the validity and reliability of a Green version of the MAQ. |
| Bramwell 2002 [54] | India, Calcutta | 127 in India | University women (India mean age = 20; U.K. mean age = 21.) | MAQ | Assess performance and factor structure of the MAQ in U.K. and Indian samples. |
| | U.K., NR. (Urban) | 112 in U.K. | | | |
| Brooks-Gunn 1980 [55] | United States, New Jersey (Suburban) | 345 female undergrads; 72 adolescents | Undergraduate women, sample of adolescent girls (6th-7th grade) | MAQ | Report the development of the Menstrual Attitudes Questionnaire (MAQ), test dimensionality and validity. |
| Chrisler 1994 [56] | United States, Connecticut | 50 Study #1; 40 Study #2 | Undergraduate women, mean age was 19 (Study #1) and 20 (Study #2) | MAQ | Studies whether the order of administration of the MAQ, MDQ, and MJQ affects responses |
| Cooper 2006 [57] | U.K. (stratified cohort) | 1050 | Baseline measurement at 14–15. Endline at 48. | Self-reported age at menarche | Compare age at menarche self-reported in adolescence to self-reported recall at age 48. |
| Darabi 2018 [58] | Iran, Tehran (Urban) | 578 | School girls ages 12–15 | The menstrual health-seeking behaviours questionnaire (MHSBQ-42) | Development and validation of a measure of menstrual health seeking behaviour informed by the theory of planned behaviour. |
| Firat 2009 [59] | Turkey (region unspecified) | 633 high school, 534 University | High school girls ages 14–18; University women ages 17–28. | MAQ | Test a modified, Turkish version of the MAQ. |
| Haver 2018 [60] | El Salvador & The Philippines | 200 (13 schools) Quantitative pilot | Pilot among menstruating girls in grades 6–8 in El Salvador | Menstrual Related–School Participation, Stress and Self-Efficacy tool (MR-SSS) | Report development of a measure of school participation, stress and self-efficacy related to menstruation. Report lessons learned, no measure results. |
| Heard 1997 [61] | United States, NR | 138 | Psychology undergraduates | The Stereotypic Beliefs About Menstrual Scale (SBAM) | Report the development of a measure of the strength and nature of beliefs in stereotypes about the menstrual cycle. |
| Jukic 2007 [62] | United States, Chicago (Urban) | 352 | Women aged (37–39) who were menstruating and not using oral contraceptives | Self-reported menstrual cycle length | Compare self-reported menstrual cycle length during a telephone survey to cycles recorded in daily diaries to assess the accuracy of cycle length reporting. |
| Khan 2017 [63] | Belize, Stann Creek | 429 households; 267 women. 17 cognitive interviews, 2 FGDs | Households and women aged 15–49; FGDs with enumerators | Self-reported menstrual hygiene (availability of private washing location, access to menstrual materials, and disposal method for materials) | Field test the performance of new WASH and menstrual hygiene management questions to monitor Sustainable Development Goal 6.1 and 6.2 targets. |
| Marvan 2006 [64] | Mexico, Puebla (Urban) | 1,090 Mexican adults (537 men, 553 women); 157 Mexican university students; 117 U.S. university students | Three samples of men and women: 1) Mexican adults ages 18–60 (male and female). 2) Mexican students age 18–24; 3) U.S. students age 18–23. | Beliefs about and Attitudes Toward Menstruation (BATM) questionnaire | Develop and test a new questionnaire for adult men and women in Mexico to assess attitudes towards menstruation. Results are compared to use of the tool in a US sample. |
| | United States, Northeast | | | | |

*(Continued)*

**Table 2.** (Continued)

| Study ID | Country, region (Urban/Rural) | Sample size (clusters) | Sample characteristics | Measure tested | Study aim |
|---|---|---|---|---|---|
| Morse 1993 [65, 66] | Canada (Urban) | 860 pre-menarche; 1,013 post-menarche | School girls in grades 6–9, ages from 10–17. | Adolescent Menstrual Attitude Questionnaire (AMAQ) | Develop and test a new measure of adolescent responses to menarche, with a premenarcheal and postmenarcheal form. |
| Ramaiya 2019 [67] | India, Uttar Pradesh (Rural) | 2,212 (240 villages), 309 in FGDs | Girls ages 12–19 participating in evaluation of an intervention. | (untitled) menstrual hygiene management scale | Develop and assess the psychometrics properties of an instrument to measure menstrual hygiene management. |
| Roberts 2004 [68] | United States, Western (Urban) | 200 | Convenience sample of premenopausal women, ages 12–61 (mean = 26). | The Menstrual Self-Evaluation Scale | Development and appraisal of the menstrual self-evaluation scale, using items from the MAQ and new items assessing self-evaluation. |
| Shin 2018 [69] | South Korea, Seoul (Urban) | 230 | School girls ages 14–19, | Menstrual Health Instrument (MHI) | Develop and test a new instrument to test menstrual health beyond clinical or premenstrual symptoms. |
| Small 2007 [70] | United States, East Coast (Urban) | 398 | Menstruating female office workers aged 19–41 not using hormonal contraception | Self-reported menstrual cycle length | To compare daily diary entries to retrospective self-reports of "usual" length of menstrual cycle on a survey |
| Stubbs 1988 [71] | United States, Boston (Suburban) | 544 | Schoolgirls pre-menarche (mean age 12.54) and post-menarche (mean age 13.88) | Menstrual Attitude Questionnaire (Adolescent) (MAQ-A) | Appraise the dimensionality and performance of the MAQ-A in a sample of adolescent girls. |
| Wegienka 2005 [72] | United States, Washington D. C. (Urban) | 385 | Premenopausal women ages 35–49 enrolled in a health plan. | Self-reported date of last menstrual period | Compared diary entries to retrospective recall of the date of onset of the last menstrual period. |
| Weller 1998 [73] | Israel, NR | 114 | First year university students, mean age 20.3 (SD = 1.21) | Self-reported menstrual regularity | Compare self-perceived menstrual 'regularity' or 'irregularity' to diary records and literature definitions of regularity. |

NR: Not Reported; WASH: Water, Sanitation and Hygiene; SD: Standard Deviation.

referred to as 'menstrual practices', 'menstrual hygiene', 'menstrual behaviours' or 'hygiene behaviours.' Eight trials included changes to menstrual practices as outcomes, making this the second most common trial outcome measure. Seven trials used menstrual and hygiene practice data to describe samples, while 5 nested studies reported on practices as part of their core research question. The list of behaviours and practices included in studies as menstrual and hygiene practices are reported in Table 5 for Review A and B studies. These do not include indexes from three studies which collapsed across an unknown set of items [20, 43, 46]. Further, in two studies, authors included attending school or university, and participating in religious practices during menstruation [16], and food restrictions and exercise [18] as menstrual practices. These fit poorly with those reported as hygiene behaviours in other studies and are not included in Table 5.

There was no consistent definition of menstrual or hygiene practices across studies, even among those using such concepts as trial outcomes. Djalalinia et al. [17] defined menstrual hygiene as *"bathing and washing during the period of menstruation after each urination and defecation, and use of sanitary pad or cotton"*, while Leventhal et al. [21] defined menstrual hygiene as the use of menstrual products and the frequency of changing products. For adolescents with mild intellectual disability, Sener and colleagues [41] evaluated menstrual hygiene as bathing practices, and observational assessment of demonstrating placing a menstrual product on a doll. Shesthra et al. [44] used menstrual practices to balance across intervention and

**Table 3. Iteratively grouped concepts measured across Review A studies by use.**

| Concepts measured | Total N | Trial outcome (primary or secondary) | Trial descriptive measure | Nested studies |
|---|---|---|---|---|
| Menstrual and hygiene practices | 19 | Dhanalakshmi 2015 [1E6] | Beksinska 2015 [14] | Hennegan 2016a [30] |
| | | Djalalinia 2012 [17] | Blake 2018 [15] | Hennegan 2016b [31] |
| | | El-Mowafy 2014 [18] | Shrestha 2018 [44] | Nyothach 2015 [35] |
| | | Fetohy 2007 [20] | Stadler 2006 [45] | Odour 2015 [36] |
| | | Leventhal 2016 [21, 23] | Wilson 2014 [48, 49] | Van Eijk 2018 [38] |
| | | Sener 2019 [41] | Womena 2018 [50] | |
| | | Sharma 2019 [43] | | |
| | | Valizadeh 2017 [46, 47] | | |
| Knowledge (menstrual and puberty) | 12 | Abedian 2011 [13] | Shrestha 2018 [44] | |
| | | Blake 2018 [15] | Womena 2018 [50] | |
| | | Dhanalakshmi 2015 [16] | | |
| | | El-Mowafy 2014 [18] | | |
| | | Fetohy 2007 [20] | | |
| | | Mbizvo 1997 [24, 25] | | |
| | | Montgomery 2016 [29] | | |
| | | Setyowati 2019 [42] | | |
| | | Sharma 2015 [43] | | |
| | | Valizadeh 2017 [46, 47] | | |
| Menstrual attitudes, beliefs, norms and restrictions | 9 | Abedian 2011 [13] | Shrestha 2018 [44] | Hennegan 2016a [30] |
| | | Blake 2018 [15] | | |
| | | Dhanalakshmi 2015 [16] | | |
| | | Djalalinia 2012 [17] | | |
| | | Fetohy 2007 [20] | | |
| | | Setyowati 2019 [42] | | |
| | | Valizadeh 2017 [46, 47] | | |
| Intervention acceptability and product preferences | 7 | Beksinska 2015 [14] | Wilson 2014 [48, 49] | Hennegan 2016a [30] |
| | | Mohamed 2014 [26] | Womena 2018 [50] | |
| | | Shrestha 2018 [44] | | |
| | | Stadler 2006 [45] | | |
| | | Womena 2018 [50] | | |
| Menstrual characteristics (including pain and symptoms) | 7 | Abedian 2011 [13] | Dhanalakshmi 2015 [16] | Van Eijk 2018 [38] |
| | | | Djalalinia 2012 [17] | Benshaul-Tolonen 2019 [40] |
| | | | Phillips-Howard 2016 [34] | |
| | | | Womena 2018 [50] | |
| Education outcomes | 7 | Montgomery 2012 [27] | | Hennegan 2016b [31] |
| | | Montgomery 2016 [29] | | Benshaul-Tolonen 2019 [40] |
| | | Oster 2011 [32] | | |
| | | Phillips-Howard 2016 [34] | | |
| | | Wilson 2014 [48, 49] | | |
| Psychosocial and well-being outcomes | 6 | Blake 2018 [15] | Phillips-Howard 2016 [34] | Hennegan 2016b [31] |
| | | Leventhal 2016 [21, 23] | | |
| | | Montgomery 2012 [27] | | |
| | | Montgomery 2016 [29] | | |
| Physical health or discomfort | 5 | Beksinska 2015 [14] | | Hennegan 2016b [31] |
| | | Phillips-Howard 2016 [34] | | |
| | | Stadler 2006 [45] | | Juma 2017 [37] |

*(Continued)*

**Table 3.** (Continued)

| Concepts measured | Total N | Trial outcome (primary or secondary) | Trial descriptive measure | Nested studies |
|---|---|---|---|---|
| Water, Sanitation and Hygiene conditions and access | 5 | | Beksinska 2015 [14] | Alexander 2018 [39] |
| | | | Blake 2018 [15] | |
| | | | Montgomery 2016 [29] | |
| | | | Phillips-Howard 2016 [34] | |
| Menstrual health | 1 | Fakhri 2012 [19] | | |
| *Other*: Sexual risk behaviours, peer product use | 1, 1 | Dhanalakshmi 2015 [16] | | Oster 2012 [33] |

control groups at baseline, implying hygiene as the menstrual materials used and if respondents had ever used sanitary pads.

In nested studies, the menstrual and hygiene behaviours of focus were defined by the study objectives. Both Nyothach et al. [35] and Oduor et al. [36] investigated hand washing before and after changing menstrual cups as 'handwashing for menstrual hygiene' and the frequency of dropping menstrual products and subsequent cleaning or management of those products, respectively. In one nested study, [31] authors based the assessment of menstrual hygiene on a pre-existing definition developed by the Joint Monitoring Programme of the World Health Organization and UNICEF in 2012 [74] to report the prevalence of the concept and its association with other outcomes.

Very few studies reported the questions used to asses menstrual and hygiene practices. Thus, it was unclear what the recall periods, question structures, and response options were for most studies. Five studies disclosed the self-report questions used for core practices assessed [21, 30, 31, 35, 36]. Among these, 'usual' practice was most commonly assessed, with some asking for ordinal responses ('always', 'sometimes', 'never') to characterise their practice. Two studies used 'this recent period' as the recall period [35, 36].

Two studies included insights on the reliability of menstrual and hygiene practices questions used. van Eijk 2018 [38] found poor agreement between self-reported menstrual cup use and cup use measured by observed change of the cup colour (kappa = 0.044). Womena 2018 [50] reported that quantitative survey responses in which respondents reported washing their materials with soap and water contradicted qualitative accounts wherein participants reported inadequate access to soap for washing.

**Review B.** Among measure development studies, two focused on measuring menstrual hygiene [63, 67]. An additional two measures included menstrual health or hygiene behaviours as subscales [51, 58]. Where reported in studies, the practices measured in these scales are included in Table 5.

*Menstrual hygiene.* As part of the development of new measures for the Multiple Indicator Cluster Surveys (MICS) [75], Khan and colleagues [63] field tested questions on menstrual hygiene. They undertook cognitive interviews with respondents and focus group discussions with interviewers administering the surveys. Results indicated that for the three menstrual questions asked, in 38–52% of cases interviewers needed to clarify the questions or probe to elicit responses. Cognitive interviews suggested that the term 'washing' may have been interpreted by respondents to infer bathing, while the term 'privacy' was understood differently among respondents in interviews. In focus groups, interviewers suggested that 'materials' was not consistently perceived to mean a menstrual absorbent.

*Menstrual hygiene management scale.* Ramaiya [67] developed a menstrual hygiene measure based on a past framework, and included domains of: use of menstrual absorbents (cloth,

**Table 4. Review B measures according to iterative concept groupings, with measure description, example items and links to concepts measured in Review A studies.**

| Measure | Original Development | Measurement construct(s) and subscales with example items | Review A concept(s) |
|---|---|---|---|
| **Attitudes, beliefs and stereotypes** | | | |
| Menstrual Attitude Questionnaire (MAQ), Menstrual Attitude Questionnaire–Adolescent Form (MAQ-A) | Brooks-Gunn 1980 (US) [55] | The questionnaire was developed to measure multidimensional menstrual-related attitudes; both positive and negative. The tool aimed *"to explore the nature of women's attitudes toward menstruation and to examine possible dimensions or styles of coping related to menstruation."* Authors also developed an adolescent-friendly version of the questions (MAQ-A). Items informed by past research, and ideas of balance across negative and positive phrasings as well as four pre-hypothesised attitude groups. The measure has five factors/sub-scales. Menstruation as: <br> 1. A debilitating event (e.g., "Avoiding certain activities during menstruation is often very wise") <br> 2. A natural event (e.g., "Menstruation provides a way for me to keep in touch with my body") <br> 3. A bothersome event (e.g., "Menstruation is something I just have to put up with") <br> 4. An event whose onset can be predicted and anticipated (e.g., "I have learned to anticipate my menstrual period by the mood changes which precede it") <br> 5. An event that does not and should not affect one's behaviour (e.g., "Premenstrual tension/irritability is all in a woman's head") <br> **Revalidation or appraisals, including new languages:** Bramwell 2002 (UK, India) [54]; Firat 2009 (Turkish, Turkey) [59]; Bargoita 2016 (Greek, Greece) [53]; Stubbs 1988 (US) [71]. **Order effects:** Aubeeluck 2002 (UK) [52]; Chrisler 1994 (US) [56] | Menstrual attitudes, beliefs, norms and restrictions |
| The Stereotypic Beliefs About Menstruation Scale (SBAM) | Heard 1977 (US)[61] | Designed to measure the strength, prevalence and nature of negative stereotypic beliefs about menstruation. [Reported in conference abstract report only]. The measure has four sub-scales: <br> 1. Danger (e.g., "Women should not hold positions of power or authority because of mood changes during the menstrual cycle") <br> 2. Stigma (e.g., "Menstruation should be kept secret") <br> 3. Superstition (e.g., "Women should not cook while menstruating") <br> 4. 4) Disability (e.g., "Menstruation is a significant cause of absence from work or school for women") | Menstrual attitudes, beliefs, norms and restrictions |
| Adolescent Menstrual Attitude Questionnaire (AMAQ) | Morse 1993 (Canada) [65, 66] | Designed to measure adolescents' attitudes towards menstruation both before and after menarche. Hypothesised domains were informed by qualitative study of adolescent reactions to menarche. Final factors (sub-scales) were: <br> 1. Positive Feelings (e.g., "I feel proud when I have my period/I will feel proud when I get my period") <br> 2. Negative Feelings (e.g., "I worry a lot about periods starting unexpectedly/I worry a lot about my periods starting") <br> 3. Living with Menstruation (e.g., "Girls with periods should avoid exercise") <br> 4. Openness (e.g., "I like to talk about periods with my friends") <br> 5. Acceptance of Menarche (e.g., "Coping with periods is easy") <br> 6. Menstrual Symptoms (e.g., "Menstruating girls are grumpy and tense") | Menstrual attitudes, beliefs, norms and restrictions |
| Beliefs about and Attitudes Toward Menstruation Questionnaire (BATM) | Marvan 2006 (Mexico) [64] | This measure was designed to assess attitudes of men and women towards menstruation. Authors designed the measure not to include any personal statements and include beliefs about activities women should and should not do while menstruating. Items were developed through review of literature on myths, stereotypes and attitudes. Initial development indicated five factors: <br> 1. Secrecy (e.g., "It is important to discuss the topic of the period at school with boys and girls together") <br> 2. Annoyance (e.g., "It is annoying for women to have the period every month") <br> 3. Proscriptions and prescriptions (e.g., "Women must avoid exercising while they are having their periods") <br> 4. Disability (e.g., "The period affects women's abilities to do housework") <br> 5. 5) Pleasant (e.g., "There are women who feel content to have their periods") | Menstrual attitudes, beliefs, norms and restrictions |
| Menstrual Self-Evaluation Scale | Roberts 2004 (US) [68] | Measure developed to measure women's attitudes and emotions towards menstruation. The measure used two subscales from the MAQ, with six new items added for this scale. Four factors (sub-scales) emerged: <br> 1. Menstruation as Bothersome (as appears in the MAQ) <br> 2. Menstruation as Disgusting or Shameful (e.g., "I would feel ashamed if I 'leaked' menstrual blood on my clothes") <br> 3. Menstruation as Enabling Awareness of One's Body (e.g., "Menstruation is a reoccurring affirmation of womanhood") <br> 4. Menstruation as Life Affirming (e.g., "The recurrent monthly flow of menstruation is an external indication of a woman's general good health") | Menstrual attitudes, beliefs, norms and restrictions |

*(Continued)*

**Table 4.** (Continued)

| Measure | Original Development | Measurement construct(s) and subscales with example items | Review A concept(s) |
|---|---|---|---|
| **Health, hygiene, self-care and help-seeking** | | | |
| Female adolescents' sexual reproductive self-care scale | Alimoradi 2019 (Iran) [51] | Designed to measure self-care in relation to sexual and reproductive health, and to capture constructs influencing reproductive and sexual self-care such as interactions with parents, knowledge and attitudes. Items were informed by a qualitative study and past research. The final tool has seven factors/sub-scales:<br>1. Adolescents and family interaction (e.g., "My father/mother welcome that I raise my issues with them")<br>2. The perception of female adolescents of premarital sexual relationships (e.g., "I think that having sexual relationships with a boyfriend is a sin")<br>3. Enabling factors for sexual and reproductive self-care (e.g., "Access to healthcare services such as visits to a gynaecologist or midwife, a psychologist and a nutritionist etc. enhances my ability for reproductive and sexual health self-care")<br>4. Understanding and behaviours of female adolescents of the interaction with the opposite sex (e.g., "Because of my adherence to family principles, I refrain from having a relationship with a boy")<br>5. Parent-adolescent communication barriers (e.g., "I do not want to talk with my mother/father about issues related to the opposite sex/puberty and menstrual cycle, so that our respect is preserved")<br>6. Reproductive and sexual knowledge (e.g., "Genital ulcers are a sign of STDs")<br>7. Self-care for reproductive health and menstruation (e.g., "If I cannot dry my underwear in the sun, I use a hot iron") | Sexual risk behaviours<br>Menstrual knowledge<br>Menstrual and hygiene practices<br>Concepts unmeasured in Review A studies |
| Menstrual health-seeking behaviours questionnaire (MHSBQ-47) | Darabi 2018 (Iran) [58] | Developed to measure menstrual health-seeking behaviours, informed by the theory of planned behaviour (TPB). Items were designed to assess constructs across the TPB for menstrual health behaviours. The measure was informed by past measures and qualitative study. Final factors (sub-scales) were:<br>1. Attitudes towards menstrual health (e.g., "Menstruation causes difficulties in concentrating on some activities such as education")<br>2. Subjective norms (e.g., "My family believes that I should continue my social activities during menstruation")<br>3. Perceived behavioural control (e.g., "I can take a shower during my menstrual period")<br>4. Perceived parental control (e.g., "My parents determine how much I should read about the puberty health-related issues")<br>5. Behavioural intention (e.g., "I have decided to frequently change my menstrual pad during my menstrual period")<br>6. Menstrual health behaviours (e.g., "I don't go to the sea and the pool during my menstrual period"; "I would use cotton underclothes during my menstrual period") | Menstrual attitudes, beliefs, norms and restrictions<br>Menstrual and hygiene practices<br>Concepts unmeasured in Review A studies |
| Menstrual hygiene management | Khan 2017 (Belize) [63] | 3 questions were tested, assessing: the availability of a private location for washing during menstruation, access to menstrual materials (and the type of menstrual materials), and the method of disposal of menstrual materials. | Menstrual practices |
| Menstrual hygiene management scale (untitled) | Ramaiya 2019 (India) [67] | A tool to measure menstrual hygiene management was designed to support evaluation of a menstrual health and hygiene intervention. Items were drawn from past definitions of menstrual hygiene, past studies and frameworks within the literature. Items largely concerned behavioural practices and menstrual environments. Following principal components analysis, authors proposed two factors/sub-scales:<br>1. Menstrual health (e.g., "What kind of menstrual absorbent do you use?")<br>2. Menstrual hygiene (e.g., "Is there a separate bathing place at home?") | Menstrual practices |
| Menstrual Health Instrument | Shin 2018 (South Korea) [69] | Designed to measure menstrual health, authors note that the measure sought to assess menstrua health more holistically beyond clinical dysmenorrhea or premenstrual symptoms. Item pool was developed through literature review. The measure includes five factors:<br>1. Affective symptoms (e.g, "I have mood swings during my period")<br>2. Somatic symptoms and school life (e.g., "I have lower abdominal pain or discomfort during my period")<br>3. Daily habits for menstrual health (e.g., "I have dietary habits of eating less salty food and taking less caffeine")<br>4. Menstrual cycle characteristics (e.g., "I have healthy menstrual cycles and periods")<br>5. Attitudes and perceptions on menstruation (e.g., "I think menstruation is an important indicator of women's overall health") | Menstrual characteristics<br>Menstrual health |

(*Continued*)

**Table 4.** (Continued)

| Measure | Original Development | Measurement construct(s) and subscales with example items | Review A concept(s) |
|---|---|---|---|
| **Impact of menstruation on school and well-being** | | | |
| Menstrual Related–School participation, stress and self-efficacy tool (MR-SSS) | Haver 2018 (Philippines; El Salvador) [60] | Developed to measure school participation, stress and self-efficacy related to menstruation. The included study describes the development of the tool but no quantitative validation results are presented. Qualitative studies informed the measure development. Authors note exploratory analyses were inconclusive. No factors or example items are reported but the paper describes three measurement domains: 1. School participation 2. Stress 3. Self-efficacy | Psychosocial and well-being outcomes Education outcomes Concepts unmeasured in Review A studies |
| **Menstrual characteristics recall** | | | |
| Age at menarche | Cooper 2006 (UK) [57] | Self-reported age at menarche at 14–15 years, compared with self-report at 48. | Menstrual characteristics |
| Menstrual cycle length | Jukic 2007 (US) [62] | Self-reported cycle length compared with mean cycle length from diary entries. | Menstrual characteristics |
| | Small 2007 (US) [70] | | |
| Date of last menstrual period | Wegienka 2005 (US) [72] | Self-reported date of last menstrual period compared with diaries. | Menstrual characteristics |
| Cycle regularity | Weller 1998 (Israel) [73] | Self-reported 'irregularity' or 'regularity' compared to diaries through which investigators defined irregularity as one third of cycles reported being less than 21 days or more than 35 days in length. | Menstrual characteristics |

Audit of measures used across included studies are reported according to the groupings in Table 3.

sanitary pad) and hygiene behaviours, which included genital washing, handwashing and bathing daily (see Table 5). Items were reported along with the categorisation of practices as 'adequate', 'semi-adequate' or 'inadequate' menstrual hygiene. Factors at baseline and endline differed with items all loading on individual factors in final analysis. The author suggested that revised understandings of the constructs were needed.

In two broad measures of 'reproductive self-care' [51] and 'menstrual health seeking behaviours' [58], sub-scales included measures of menstrual self-care practices. The former included the subscale 'Self-care for reproductive health and menstruation', this included the menstrual practices listed in Table 5, along with items that fit more poorly with menstrual hygiene practices reported in most Review A studies including: using iron pills, recording the dates of the menstrual period, using painkillers to manage menstrual pain, and monitoring menstrual blood loss. The latter measure assessed menstrual health seeking across constructs from the theory of planned behaviour (TPB) and included a subscale on behaviours. Alongside the practices listed in Table 5, authors included items reported to capture menstrual health behaviours including; avoiding swimming in pools during menses, avoiding caffeine and reducing aggression during menstruation. These items evidenced inconsistencies in the boundaries of self-care behaviours, restrictions during menstruation, and hygiene practices.

## Knowledge

**Review A.** Knowledge about menstruation and puberty was the most frequent outcome assessed in trials (n = 10). A further two studies reported measuring menstrual knowledge for descriptive purposes. Knowledge was assessed through tests.

The content of knowledge assessments varied, as did the level of detail provided by authors about the topics covered. Although noting expert or text-book input into knowledge measures, four studies provided no information on the content included in the tests [13, 16, 42, 43]. For

**Table 5. Menstrual and hygiene practices measured across included studies.**

| Menstrual and hygiene practices | N | Review A Studies | N | Review B Measures |
|---|---|---|---|---|
| Menstrual product used | 15 | Beksinska 2015; Blake 2018; Dhanalakshmi 2015 Djalalinia 2012; El-Mowafy 2014; Leventhal 2016; Hennegan 2016a; Hennegan 2016b; Nyothach 2015 | 2 | Khan 2017; Ramaiya 2019 |
| | | Odour 2015; Van Eijk 2018; Shestha 2018; Stadler 2006; Wilson 2014; Womena 2018 | | |
| Frequency of changing menstrual products | 5 | El-Mowafy 2014; Leventhal 2016; Hennegan 2016b; Sharma 2019; Womena 2018 | 2 | Ramaiya 2019; Alimoradi 2019 |
| Number of products used/day | 4 | Dhanalakshmi 2015; El-Mowafy 2014; Hennegan 2016a; Stadler 2006 | | |
| Disposal location (at home/ at school/ unspecified) | 4 | Beksinska 2015 (home); Blake 2018 (school); El-Mowafy 2014 (unspecified); Hennegan 2016b (school) | 2 | Khan 2017 (unspecified); Ramaiya 2019 (unspecified); |
| Drying method/location | 4 | El-Mowafy 2014; Hennegan 2016b; Wilson 2014; Womena 2018 | 1 | Ramaiya 2019 |
| Washing materials with soap | 3 | Hennegan 2016b; Sharma 2019; Wilson 2014 | 1 | Ramaiya 2019 |
| Bathing during menstruation (at all or frequency) | 3 | Dhanalakshmi 2015; Djalalinia 2012; Sener 2019 | 1 | Ramaiya 2019 |
| Location of changing menstrual materials (home/school/unspecified) | 2 | Hennegan 2016b; Womena 2018 | | |
| Handwashing before changing/emptying product | 2 | Nyothach 2015; Womena 2018 | | |
| Handwashing after changing/emptying product | 2 | Nyothach 2015; Womena 2018 | | |
| Perineal care | 2 | Dhanalakshmi 2015; El-Mowafy 2014 | | |
| Ever used a sanitary pad | 1 | Shestha 2018 | | |
| Storage of menstrual products | 1 | Womena 2018 | | Ramaiya 2019 |
| Ability to place menstrual product (demonstrated on doll) | 1 | Sener 2019 | | |
| Access to underwear | 1 | Womena 2018 | | |
| Sharing cloths with others | 1 | Hennegan 2016a | | |
| Dropping material | 1 | Odour 2015 | | |
| Location of dropping (home school) | 1 | Odour 2015 | | |
| Cleaning dropped materials | 1 | Odour 2015 | | |
| Washing after each urination and defecation | 1 | Djalalinia 2012 | | |
| Cleanliness of genitalia | 1 | Sharma 2019 | | |
| Basin to wash materials | 1 | Womena 2018 | | |
| Washing location | 1 | Womena 2018 | | |
| Privacy concerns during washing materials | 1 | Hennegan 2016b | | |
| Washing menstrual materials (with other materials or alone) | 1 | El-Mowafy 2014 | | |
| Type of water used for washing (hot/ cold) | 1 | El-Mowafy 2014 | | |
| Underwear (methods of cleaning) | 1 | El-Mowafy 2014 | | |
| Drying covered/not covered | 1 | Womena 2018 | | |
| Frequency of wearing materials wet | 1 | Hennegan 2016b | | |
| Boiling menstrual cup | 1 | Womena 2018 | | |
| Access to pain relief | 1 | Womena 2018 | | |
| Method of shaving hair in genital area | 1 | El-Mowafy 2014 | | |
| Cleaning genitals after each bowel movement | 0 | | 2 | Alimoradi 2019; Darabi 2018 |
| Changing underwear daily | 0 | | 2 | Alimoradi 2019; Darabi 2018 |
| Frequency of disposal | 0 | | 1 | Ramaiya 2019 |

*(Continued)*

**Table 5.** (Continued)

| Menstrual and hygiene practices | N | Review A Studies | N | Review B Measures |
|---|---|---|---|---|
| Cleaning genitals with each change of menstrual material | 0 | | 1 | Ramaiya 2019 |
| Privacy for washing the body | 0 | | 1 | Khan 2017 |
| Privacy for changing materials | 0 | | 1 | Ramaiya 2019 |
| Bathing standing during menstruation | 0 | | 1 | Darabi 2018 |
| Handwashing with soap before cleaning genitals | 0 | | 1 | Ramaiya 2019 |
| Handwashing with soap after cleaning genitals | 0 | | 1 | Ramaiya 2019 |
| Washing underwear | 0 | | 1 | Alimoradi 2019 |
| Ironing underwear or drying in the sun | 0 | | 1 | Alimoradi 2019 |
| Wearing breathable clothes or underwear | 0 | | 1 | Darabi 2018 |

the studies that did provide example questions or indicate the coverage of knowledge content, there was a wide range. The indication that menstruation was a physical process, the age of onset of menarche, and the origin of menstrual blood as the female reproductive tract (uterus, through the vagina) were common across studies. Some studies broadened biological knowledge to include secondary sexual characteristic changes during puberty such as the development of breasts or hips, the timing of ovulation and links between menstrual cycle and reproduction. Others included knowledge of menstrual disorders e.g., the definition of dysmenorrhea, causes of pain or discomfort, and included questions about self-care for pain during menses [18, 20]. Four studies reported including questions about hygiene practices during menstruation as part of knowledge assessments such as the types of materials to use as absorbents, and the frequency with which one should change materials [20, 24, 44, 46].

In their list of examples, Blake 2018 [15] included "Girls should stay home from school when they are menstruating" as part of their knowledge test as a reflection of the content of the puberty book provided as part of the intervention. A second study [46] also stated including questions about avoiding foods or physical activity practices to be undertaken during menses in knowledge assessments. These questions were similar to those used to capture menstrual restrictions in other studies.

**Review B.**   We did not identify any eligible studies developing measures of menstrual knowledge. One broad measure of reproductive self-care included a subscale on knowledge and attitudes towards open discussion of sexual and reproductive health topics [51]. In this sub-scale, some items concerned freedom to discuss menstrual and reproductive health topics with parents, although many items were more focused on sexual knowledge such as sexually transmitted diseases.

## Intervention acceptability and product preferences

**Review A.**   In our iterative groupings, eight studies included measures aiming to capture the acceptability of the tested interventions. In three studies comparing different menstrual products, satisfaction, comfort and product preferences or willingness to continue use were primary outcomes [14, 26, 45]. Mohamed and colleagues 2014, [26] also compared physiological responses to wearing sanitary pads of different thicknesses to further assess product performance. Shrestha 2018 [44] compared product demand in response to awareness raising and

discounts, using coupon redemption for products as an objective measure of demand. Two other pilot trials sought to understand the acceptability of interventions to their target recipients using qualitative feedback and self-report [48, 50]. One nested study compared ratings of product reliability, comfort and satisfaction between participants given a reusable sanitary pad as part of the trial to those using their existing materials, and assessed participant willingness to continue use of the provided pad [30]. Questions used across studies varied specific to research questions or intervention.

**Review B.**   No included studies developed measures of menstrual product or intervention acceptability.

## Menstrual attitudes, beliefs, norms and restrictions

**Review A.**   Five trials assessed intervention impacts on menstrual attitudes, and one additional trial reported assessing 'menarche experience' comparing self-reported attitudes at menarche. Two studies used the MAQ [55] and AMAQ [65, 66]. Three studies used self-created attitudes questionnaires. Blake et al. [15] described a combined knowledge and attitudes questionnaire, although it was unclear what attitude questions were included and no examples were provided. Additionally, the study included two items capturing fear and shame in association with menstruating, with the items reported in full: "Does the idea of menstruating make you feel afraid or shameful?" and "Is talking about menstruation something that is shameful for you?". Similar items were described as psychosocial well-being outcomes in another program of work [27, 29]. Fetohy [20] provided little description of the menstrual attitude scale used, noting only that it assessed "attitude toward healthy and unhealthy practices during menstruation". Afsari and colleagues [47] developed a 15-question attitude scale which included items regarding menstruation, nutrition, exercise, physical activity and psychological and mental health, but no further detail was provided. Djalalinia [17] reported comparing whether participants reported feeling good (happy or proud, and thankful) at the moment of menarche in methods, and in results presented if participants felt confused, scared, uncomfortable, or good.

One study assessed the impact of a menstrual intervention on beliefs regarding menstruation and sexual behaviour. In the study, this was separated from menstrual knowledge and included ten items capturing: whether girls should continue education after menarche, marriageability at menarche, and if there was a relationship between eating sweets and menstrual bleeding. A second study [44] assessed stigma and norms about menstruation, although it did not feature in the main text reporting of results. Stigma questions were described and asked women to report whether they were allowed in the kitchen, holy places, secluded to a shed or considered untouchable during menstruation; these were more similar to items capturing restrictions in another study. As noted in the section on menstrual and hygiene practices, some studies included avoiding physical activity, education or other activities during menstruation as practices. One nested study [30] compared the impact of menstrual materials used by participants on whether they avoided daily activities such as cooking, or doing sports during menstruation, and if there were activities menstruation caused them to miss. These were compared according to the menstrual materials used suggesting they reflected limitations due to material quality, but these also may reflect attitudes towards appropriate behaviours during menstruation and did not fit easily into any iterative groupings in the review.

**Review B.**   Most Review B measure development studies sought to capture attitudes towards menstruation (Table 4).

*The Menstrual Attitude Questionnaire (MAQ).* The MAQ was developed in 1980. The initial questionnaire was tested among both male and female university students. Concurrently,

developers proposed a shorter, simplified version for adolescent individuals pre and post menarche. As displayed in Table 3, the scale has five factors.

Attempts to revalidate the MAQ produced mixed results. Stubbs and colleagues (1988) found poor internal consistency of subscales and proposed an alternate factor structure, capturing 'affirmations' of menstruation, and 'worry' or dislike of menstruation. Authors assess validation through relationships with depression and anxiety symptoms, self-esteem, locus of control, and body satisfaction. Over a decade later, Bramwell and colleagues (2002) found the MAQ factor structure was not an acceptable fit in British or Indian samples. Similarly, a Turkish version of the MAQ exhibited poor fit for the original factor structure. Authors undertook follow-up exploratory analyses to propose an alternate factor structure broadly like the original MAQ, with some items loaded differently. The original factor structure was not supported in a Greek version of the MAQ, although similar to the Turkish version, follow-up EFA determined a similar five-factors with items re-distributed.

With the hypothesis that the framing of menstruation in a positive rather than negative light might cause respondents to view their menstrual experiences differently, two studies [52, 56] used a parody Menstrual Joy Questionnaire (MJQ) (not eligible for this review due to a focus on symptoms) to examine the effect of priming positive views of menstruation on responses to the MAQ, as well as a questionnaire assessing the severity of menstrual symptoms not eligible for this review (the Menstrual Distress Questionnaire, MDQ). Chrisler et al.[56] found exposure to the MJQ resulted in more positive responses to menstruation on the MAQ, while in a second study Aubeeluck et al. [52] found that the MAQ primarily increased scores on the 'menstruation as a natural event' subscale of the MAQ.

*The Adolescent Menstrual Attitude Questionnaire (AMAQ)*. In 1993, Morse and colleagues developed an alternate attitude measure for adolescents with a form for premenarcheal and postmenarcheal girls. Developed from the results of qualitative studies rather than the MAQ, the AMAQ covers a different range of topics (see Table 4). Authors sought to identify differences between girls before and after menarche and develop a meaningful measure for both groups.

*Stereotypic Beliefs about Menstruation Scale (SBAM)*. This measure was reported only in a conference abstract and focused on the assessment of negative stereotypical beliefs about menstruation [61]. Authors compared subscale scores between men and women (see Table 4).

*Beliefs about and attitudes towards menstruation questionnaire (BATM)*. Marvan and colleagues [64] developed the BATM to capture Mexican adults' attitudes towards menstruation (Table 4). The measure was tested among Mexican and North American populations. Authors noted differences between samples with the Mexican sample reporting higher expectations for avoiding swimming, carrying heavy items and avoiding certain foods. Men also endorsed higher proscriptive attitudes than women.

*Menstrual self-evaluation scale*. This measure was developed in 2004 and uses two subscales from the MAQ ('bothersome' and 'menstruation as a natural event') in addition to six new items [68]. Self-evaluation scale scores were associated with self-objectification and objectified body consciousness, consistent with author hypotheses as predictive validity.

## Menstrual characteristics

**Review A.**   One trial (Abedian) measured menstrual pain and blood loss using a variety of measures beyond the scope of this review to compare between intervention education conditions and controls. Four other trials used a variety of menstrual characteristics to provide a picture of the sample [16, 17, 34, 50]. All described the proportion of the population experiencing menstrual pain or dysmenorrhea. This was self-reported, although no studies described the

questions used. Three studies also reported the duration of menses, either in the average number of days [34, 50] or the proportion of the sample experiencing 'longer periods' defined as more than 5 days of bleeding [16]. Age at menarche was reported by Phillips-Howard et al. [34], and used in one nested study to compare menstrual cup uptake according to the time since menarche [38].

**Review B.** Five measure development studies focused on the accuracy of self-reported menstrual characteristics. These studies evaluated the timing and modality of surveys and their impact on recall accuracy. Two studies examined women's self-reported cycle length, comparing respondents cycle recorded in daily diaries to self-reported average cycle length [62, 70]. Jukic [62] reported that diary recorded cycle length over 6 months showed moderate agreement with self-reported cycle length (kappa = 0.45) and reported that women overestimated their cycle length by an average 0.7 days. In this study, almost 35% of women reported having a 28-day cycle, but the diary observed cycles suggested a less peaked distribution with between 15 and 20% of women recording cycles 26–28 days. In contrast, Small [70] found a similar near 40% of women reporting a 28-day cycle, while diary data suggested a less peaked distribution. In this study women underestimated their cycle length by 1.5 days.

Weller [73] assessed women's concepts of menstrual regularity and irregularity, comparing self-reported irregularity to diary records of menses. Authors coded irregularity as those with one third of cycles over a six-month period as more than 35 days or less than 21. Approximately 70% were classified as regular in both methods. However, only 44% who reported irregularity were coded as irregular using diary data, while some of the women who considered their periods to be regular were coded to have irregular periods (18%).

Wegienka [72] found that 56% of women accurately reported the date of their last menstrual period and that 81% reported it accurately within +/- two days. Authors also noted that a duration of 3 weeks or longer since the last period was associated with overestimation of the time since the last period. In their US sample, education level was not associated with recall accuracy. Cooper [57] found that 85% of women at age 48 accurately recalled their age at menarche (reported when 14–15 years old) within 1 year.

## Education outcomes

Multiple trials assessed the impact of MH interventions on education. School attendance measured through a combination of school registers [27, 29, 40], spot checks [29, 40], diaries [32, 34] and self-report surveys [48] was the main education outcome measured in all studies assessing this concept. Phillips-Howard et al. [34] also assessed school drop out as a primary outcome. Attendance across all days was used in both Montgomery 2012 and 2016 [27, 29], while Oster 2011 used a combination of attendance registers, absenteeism diaries and menstrual records to assess absences during menstruation [32]. Phillips-Howard et al. [34] found that diaries were unreliable and were unable to compare conditions on attendance using this method and so did not report findings in the primary trial report. In 2019, Benshaul-Tolonen and colleagues used the school register and spot-check data collected as part of that trial to compare intervention effects [40]. This study also used additional register and spot-check data on non-trial students and boys in the trial schools to compare attendance as recorded by these different methods, finding non-random inconsistencies between spot-checks and attendance records. Wilson et al. [48] reported descriptively the days missed due to menstruation but compared conditions based on the total number of days girls self-reported missing in the preceding month.

Educational engagement was inferred in a pilot trial and one nested study [27, 31]. In surveys, girls self-reported if they were able to concentrate in school during their period. Identified through Review B, Haver and colleagues described efforts to develop a measure including

menstrual related self-efficacy, stress and school participation [60]. They did not report tests of a final measure, discussing difficulties in initial piloting and validation. Authors noted that items used to measure the three pre-defined latent constructs overlapped in girls' experiences.

## Psychosocial and well-being outcomes

**Review A.**   Two trials compared scores on the Strengths and Difficulties Questionnaire [76] as secondary psychosocial well-being outcome [27, 29], and this score was also included in study nested in the second trial [31]. In addition, these trials included items asking girls to report on shame and insecurity during menstruation compared to when they were not menstruating. These items are similar to those assessed by Blake et al. [15] which assessed girls fear and shame associated with menstruation. Leventhal et al. [21] included a range of psychosocial measures as outcomes, although linked these to the resiliency components of the intervention rather than any attention to menstruation [22]. These measures included emotional resiliency, general self-efficacy, social-emotional assets, depression, general anxiety, positive psychological well-being and social well-being; all using previously developed and established measures [23]. Phillips-Howard et al. [34] reported assessing wellbeing through the Paediatric Quality of Life Inventory (PEDSQL; [77]) and used scores to describe balance across trial conditions at baseline. Well-being was not compared as an outcome in trial reports, although was listed among secondary outcomes in trial registration.

**Review B.**   The inclusion criteria for this review were designed to target menstrual-specific measures. Generalised measures of psychosocial functioning and well-being are beyond the scope of this review. One included measure study described the development of a measure of menstrual-related psychosocial outcomes including sub-scales on stress and self-efficacy, but noted that these concepts overlapped in pilot testing and did not report a final measure [60].

## Physical health or discomfort

Two trials comparing menstrual products assessed discomfort. Beksinska et al. [14] asked participants to report comfort levels and adverse events associated with the sanitary pads and menstrual cups, including vaginal irritation and dysuria. Stalder et al. [45] asked participants to report on discomfort attributes including; feeling hot/sweaty/stuffy, itching, chafing, soreness/tenderness, redness or presence of a rash, wet or sticking feeling, or burning. The Phillips-Howard trial [34] compared the impact of sanitary pad and menstrual cup provision on sexually transmitted diseases, *C. trachomatis*, *T. vaginalis*, and *N. gonorrhoea*, and reproductive tract infections bacterial vaginosis or *C. albicans* using vaginal self-swabs. They also tested for adverse events, including the presence of *Staphylococcus aureus*. A study nested within this trial further explored potential for adverse events by testing the presence of *S. aureus* vaginal colonization and *Escherichia coli* grown on samples of menstrual cups provided to girls [37]. These were compared according to the duration of cup use.

One nested study investigated the association of menstrual hygiene practices reported by participants in the trial with self-reported reproductive tract infection symptoms which included: skin irritation or rashes in the pelvic area, itching or burning in the pelvic area, and white or green vaginal discharge since the start of the school year [31].

Reproductive tract infection and biomarker methods were beyond the scope of our Review B inclusion criteria focused on menstrual experience.

## WASH

In five studies, investigators assessed Water, Sanitation and Hygiene (WASH) conditions. WASH conditions were assessed to describe the comparability of infrastructure between

eligible schools and intervention conditions. In one trial, self-reported WASH infrastructure at home was used to screen for eligible participants (those with a municipal water supply). In one nested study, investigators assessed if there were any changes in WASH conditions throughout the duration of the study; infrastructure to support menstruation did not improve, although there was increased soap availability [39].

## Menstrual health

**Review A.** One study used a combined measure of 'menstrual health' as the primary outcome [19]. This included items on a range of different features of menstrual experiences, including menstrual practices, impacts of menstruation on daily activities and school attention, dietary choices, iron supplementation and exercise.

**Review B.** Two included measure studies also concerned broad conceptualisation of menstrual health.

*Menstrual Health Instrument.* Developed in a South Korean context [69], the construct was informed by literature review and authors priorities to include both symptoms experienced during the menstrual cycle as well as self-care, attitudes and perceptions about menstruation. Authors validated the measure against self-reported menstrual cycle symptoms and quality of life measures.

*Menstrual health seeking behaviours questionnaire.* Developed in Iran among adolescent girls,[58] items were based on the theory of planned behaviour (TPB) and included all TPB model components from perceived control and attitudes through to intentions and behaviours. Items were drawn from review, qualitative studies and theoretical constructs.

# Discussion

Through two linked systematic reviews we audited the measures used across trials and nested studies of menstrual health and hygiene interventions in LMICs, and measures developed across all countries to assess females' experiences of menstruation. The key finding from our audit is one of inconsistency across studies, supporting calls for greater attention to measurement [3, 7, 78]. Results indicate that to improve measurement in menstrual health and hygiene research, researchers must attend to the definition of core concepts, followed by the way these are operationalised.

## Concept definitions

By iteratively grouping measures we found that many studies assessed similar concepts, but that these were defined differently in every study or lacked clear definitions entirely. As a result, measures also differed, hampering comparisons across the evidence base. Particularly problematic concepts were menstrual and hygiene practices, menstrual knowledge, and menstrual attitudes, norms, beliefs and restrictions.

## Menstrual and hygiene practices

To improve future research our review findings suggest that: (1) if interventions continue to aim to improve 'menstrual hygiene' or 'hygiene behaviours', these concepts need to be consistently defined and operationalised for measurement, and (2) researchers should identify a core set of menstrual or hygiene practices needed to describe populations across studies.

Our audit suggests that greater attention to conceptual and definitional consistency will be key to improving quantitative research in menstrual health and hygiene. Many trials and nested studies included measures of the different practices or behaviours that participants

undertook to care for their bodies during menstruation. Forty-four different menstrual or hygiene practices were reported, with no two studies assessing the same set of practices. Few studies acknowledged this lack of consistency with other research and often used unified terms such as 'menstrual hygiene', 'hygiene behaviours' or 'menstrual practices' to label the construct measured. Recent measure development efforts seeking to assess menstrual hygiene were also inconsistent in identifying the boundaries of this construct and highlighted challenges in operationalising current definitions for measurement. Menstrual hygiene and similar concepts were frequently used as trial outcomes, or to describe study populations. Inconsistency across outcome measures means that intervention effects on menstrual hygiene, or other iterations of this term, are not comparable across trials. Inconsistency in describing population self-care practices makes it difficult to consider the generalisability of study findings. Intervention effects may not generalise to populations with different practice profiles and external validity will be difficult to appraise without consistent descriptions across studies (for example, sanitary pad provision interventions are likely to have less impact in populations already using these products).

While challenging to institute in practice, the present study suggests that identification of a core set of practices for assessment is likely of considerable value. Our audit reveals that practices measured across studies have been highly inconsistent and seemingly haphazard. That is not to say that the broad scope is a problem in-and-of-itself: some questions are relevant only for specific research questions or cultural contexts, and there can be much to be learned by examining a wide range of practices. However, until a core set of measures start to be regularly included in studies of this topic, comparability across studies and across cultural contexts will be necessarily limited.

## Menstrual knowledge

Similarly, this audit suggests that measures of knowledge are likely to be limited and inconsistent without improved clarity in concept definitions. Menstrual knowledge was the primary outcome of interest in many trials of menstrual education interventions. The coverage of topics in knowledge tests varied significantly across studies, despite most studies focusing on adolescent girls. This reflected the aims and content of the different education interventions, with researchers defining menstrual knowledge gains based on the retention of information provided in their own programs. The variation in knowledge assessed makes it difficult to compare results across studies. Improved knowledge regarding the basic biology of menstruation (origin of bleeding, anatomy of the female reproductive system, and age of menarche) may require a different intensity, duration or format of education package than what may be needed to improve knowledge about sexual health, menstrual disorders, reproductive tract infection symptoms or menstrual hygiene behaviours. This is important when comparing intervention effectiveness, and in considering the impacts of education interventions on more distal outcomes, such as health and education.

In undertaking needs assessments and evaluating the generalisability of trial results to other populations, the results of this audit suggest it may be helpful to establish different components of menstrual knowledge and their relation to the needs and experiences of women and girls. Knowledge on different topics may show different relationships with well-being and education outcomes; similarly different knowledge may be more beneficial at different life stages [79]. For example, information on anatomy may dispel fears of illness or distress when received before menarche, while more education on menstrual and pain management strategies may facilitate school engagement among those post-menarche. This comes together to suggest it may be useful to define a set of core menstrual knowledge items to be able to separate these

results from other hygiene or sexual health knowledge indicators and aid comparison across future studies.

## Attitudes, norms, beliefs and restrictions

Attitudes, norms, beliefs, and restrictions featured prominently among the questions asked in the studies we examined; however, as with many other topic areas, the ability of findings to speak to each other across the field—and to be consistent with studies in other related fields—will continue to be limited until terms are defined and deployed with more consistency. Our findings show that authors across studies used terms such as menstrual stigma, norms, restrictions, and behavioural proscriptions in varied ways, suggesting a lack of clarity around these concepts. These terminologies were used across studies with overlapping meanings and measures, and with limited reference to overarching theories such as social norm theory (e.g., [80, 81]) or mid-level theory of menstrual experience [2]. Across measure development studies this manifested in the absence of a clear nomological network. This meant few studies were able to validate scales against theoretically related constructs and assess convergent, divergent or predictive validity.

## Impact measures

Continued efforts to identify and use standardised measures when testing the impact of menstrual health and hygiene interventions on non-menstrual outcomes are needed. Interventions have been hypothesised to be relevant for health, education, psychosocial well-being, gender and water, sanitation and hygiene domains [78]. Standardised measures for these outcomes were beyond the scope of this review, but were addressed in an interdisciplinary meeting to set priority impact measures for monitoring menstrual health and hygiene improvements [78]. This meeting recommended a range of potential impact measures, few of which were used in past trials identified by this review.

There continues to be a need for attention to measures of school attendance within the field. Effects on education were most commonly tested by trials in this review, with studies focused almost exclusively on attendance. Different measures including spot-checks, diaries, self-report and school attendance records were used to capture attendance. While methods are likely to suffer from different levels of bias in different contexts [40], users of education-focused outcome variables should be also be mindful of whether effect sizes reported are for outcomes specifically during menstrual periods or general absenteeism. Absences due to menstruation occur only during that phase of the menstrual cycle, reflecting a proportion of the total school days in a month. Measures of general attendance may be easier to obtain and avoid the difficulties of recording menstrual cycles, however, effect sizes will be diluted by non-menstrual days recorded.

## Operationalising measures

Findings from our review and audit provide considerations for improving the operationalisation of core concepts into valid and reliable measures in the study of menstrual health and hygiene.

To improve measurement across research, all studies should clearly report the questions used. Full disclosure of questionnaires through online supplementary materials and alongside trial registrations would significantly improve readers' and reviewers' ability to compare findings across studies. This would also facilitate transparency and allow critical appraisal of the questions used. Our audit was consistently limited by the unavailability of measures and limited reflection by study authors on the way core concepts were operationalised into questions.

This audit reveals that there is far more opportunity for questionnaire validation within the field. Few measures used in trials had been formally validated, and only one developed measure (the MAQ [55]) had been tested beyond its original context, and performed poorly across contexts [54, 59]. Further, development and validation of measures of menstrual attitudes have also suggested significant variation in attitudes by culture [54, 64], and age [65], meaning care and cross-validation will be needed in any attempt to develop questionnaires on this domain. These findings reinforce the need for more attention and funding to the development and revalidation of measures of core concepts to advance menstrual health and hygiene research.

Similarly, the field would benefit from more attention to measure reliability, particularly since the relatively few studies focused on reliability analysis revealed some important findings. Few studies provided information on the reliability of self-reported questions. Among trial studies, two found poor agreement between self-reported menstrual practices and observational data or qualitative study findings [38, 50]. While observations are invasive and may not be feasible for many menstrual behaviours, comparison of self-report in surveys to diary entries or test-retest reliability assessments may help to assess the accuracy of current survey data and refine question formats and enumerator training. Self-reported menstrual characteristics such as the timing of menarche and the last menstrual period were found to be reliable in studies in high income countries. Less reliable were women's reports of their cycle regularity compared to author definitions, although error in self-reported cycle lengths was less than two days. These findings provide some confidence in the potential of self-report questions, but more research in LMIC contexts is needed.

In considering the reliability of self-reported menstrual experience, evidence from high income contexts suggests that priming and question order effects may need to be considered. Two studies found that priming positive menstrual expectations impacted on participants self-reported menstrual attitudes [52, 56]. This may be relevant to self-reports in intervention studies, where it is unclear how priming or questions frames focused on negative experiences of menstruation, may influence self-reported outcomes. It may be useful for developed measures to attend to balance in positively and negatively framed questions.

## Strengths and limitations of the review

Through systematic searching we provide a comprehensive picture of measures used in trials of menstrual health and hygiene interventions and measures developed to capture menstrual experiences. Our focus on trials rather than other quantitative study designs may have excluded some measures. Based on reviews of past literature [2, 5, 6], we determined that an in-depth analysis of trials and nested studies offered a more manageable set of higher quality designs in which to explore the measures used. Further, intervention trials are likely to be the highest priority for policy makers and practitioners seeking evidence to inform programming. Our searching strategy for Review B of measure studies focused on development and validation studies and may have missed studies where measures were used and appraised as part of a study with a research question not focused on measure reliability or validity.

Iterative category groupings were based on available studies and may have been constructed differently by an alternative author group. Groupings reflect the current set of studies and would likely change given a more nuanced set of included measures; for instance, improved clarity in distinguishing attitudes towards menstruation from stigma or norms around menstruation would result in more distinct groupings for these measures. In a deviation from the protocol, we did not undertake formal quality appraisal of measurement validation studies. Through auditing measured used, we found that conceptual ambiguity was the most significant concern. Positing that the quality of measures is only meaningful where they capture

relevant concepts, we dedicated our attention to concept analysis. Publication bias may mean that unsuccessful measure development efforts were not reported.

We excluded studies which developed measures to identify menstrual disorders or symptoms. This best aligned with our focus on improved measures in the study of menstrual health and hygiene interventions in LMICs. These inclusion criteria meant we excluded the menstrual distress questionnaire (MDQ) [82]. The MDQ has been used in association with the development of other menstrual experience measures such as the MAQ but focuses on physical symptoms associated with the menstrual cycle. As more attention is given to the influence of menstrual symptoms and disorders in LMICs, these measures will also need greater consideration.

## Conclusions

Through this systematic review and audit, we provide a reference point capturing the current measures used in menstrual health and hygiene research. We found that measures were inconsistent in conceptualisation and operationalisation across studies. Findings suggest that interdisciplinary efforts are needed to better define core constructs such as menstrual and hygiene practices, menstrual knowledge, attitudes, norms and restrictions. In absence of consensus among researchers, authors should clearly define the concepts measured for the purposes of their study and should transparently report measures used. We recommend including full surveys in published supplementary materials where feasible. Research is needed to develop and test reliable and valid measures for core concepts. To support measure development, mid-level theory of menstrual experiences [2] and intervention effects [78] should be used to inform priority measures and identify concepts against which such measures can be tested for validity.

## Supporting information

**S1 Table. PRISMA checklist.**
(PDF)

**S2 Table. Review A: Author reported concepts measured in trials and nested studies, assessment modality and development.**
(PDF)

**S3 Table. Review B: Development, reliability and validity of scale measures.**
(PDF)

## Acknowledgments

We gratefully acknowledge Alexandra K. Shannon who undertook the initial systematic searches and double screened titles and abstracts. We also thank Andrew M. Culver who provided additional checks of the accuracy of information contained in study tables against the included studies.

## Author Contributions

**Conceptualization:** Julie Hennegan, G. J. Melendez-Torres.

**Data curation:** Julie Hennegan.

**Formal analysis:** Julie Hennegan, Deborah Jordan Brooks, G. J. Melendez-Torres.

**Funding acquisition:** Julie Hennegan, Kellogg J. Schwab.

**Investigation:** Julie Hennegan, Deborah Jordan Brooks, Kellogg J. Schwab, G. J. Melendez-Torres.

**Methodology:** Julie Hennegan, G. J. Melendez-Torres.

**Project administration:** Julie Hennegan, Deborah Jordan Brooks, Kellogg J. Schwab.

**Supervision:** Kellogg J. Schwab.

**Writing – original draft:** Julie Hennegan, Deborah Jordan Brooks.

**Writing – review & editing:** Deborah Jordan Brooks, Kellogg J. Schwab, G. J. Melendez-Torres.

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
