## [Decision Letter · Decision Letter 0]

27 Apr 2020

Measurement in the study of menstrual health and hygiene: A systematic review and audit

PONE-D-19-30808

Dear Dr. Hennegan,

We are pleased to inform you that your manuscript has been judged scientifically suitable for publication and will be formally accepted for publication once it complies with all outstanding technical requirements.

With kind regards,

Nitika Pant Pai, MD., MPH., PhD

Academic Editor

PLOS ONE

2.  In the Methods, please describe how risk of bias was assessed in individual studies (including specification of whether this was done at the study or outcome level, or both, and the specific test employed, such as the I^2 statistic), and how this information was used in any data synthesis.

3. In the Methods, please specify any assessment of risk of bias that may affect the cumulative evidence (e.g., publication bias, selective reporting within studies). Please ensure that the specific method of assessment (funnel plot, Egger's test, Begg's test, etc) is mentioned.

Additional Editor Comments (optional):

This is a seminal review in the field of menstrual hygiene & health. The authors have elegantly conducted two systematic reviews- one, that focusses on clinical trials, and various outcomes documented therein, reflecting heterogeneity across studies. Two, a second review that documents the concepts and measures across studies. As Measures varied across studies, so did the practices and outcomes assessed therein. This review captures them elegantly.

It highlights the need for a framework that will outline the measures and metrics captured across various contexts, in the field of MHH and subtly underpinning the need for consistency.

There is a huge need to begin discussions on documentation of outcomes that are relevant and measurable and qualitative outcomes that are needed in studies /trials in this area.

So, this will be a good step in that direction.

Excellent work summarizing these concepts and divergent methods captured across studies.

Please address minor comments from reviewers 1 in your revised final manuscript.

Reviewers' comments:

Reviewer's Responses to Questions

**Comments to the Author**

1. Is the manuscript technically sound, and do the data support the conclusions?

Reviewer #1: Yes

Reviewer #2: Yes

2. Has the statistical analysis been performed appropriately and rigorously? 

Reviewer #1: N/A

Reviewer #2: Yes

3. Have the authors made all data underlying the findings in their manuscript fully available?

Reviewer #1: Yes

Reviewer #2: Yes

4. Is the manuscript presented in an intelligible fashion and written in standard English?

Reviewer #1: Yes

Reviewer #2: Yes

5. Review Comments to the Author

Reviewer #1: Thank you for the opportunity to review your paper. I found it clear and insightful; it fills a gigantic gap in the MH sector. The depth of the review and assessment of available measures is incredible, and this will be a great resource for the sector. Please find below a small number of suggested changes for your consideration:

• Page 20: Ramaiya 2019 – ‘past definitions’. Which past definitions does this refer to? The JMP 2012 MHM definition? Consider clarifying and mentioning the JMP definition (here or elsewhere) as I’ve seen that pop up in a number of MHM-related studies and was surprised to not see mention of it here in the discussion on definitions. I know the JMP definition has been used (and critiqued) elsewhere, but I wonder if it might be worth a brief mention of where/how that fits into the review since (while not a perfect definition) this is one of the first examples of agreement on an MH-related definition (at least at the global level) and could show that your recommendations are feasible and could build on efforts such as this (and the collective’s definition of MH).

• There is a lot of information here and it seems like this could be two papers. I can see an argument for combining them - it provides a single, more comprehensive resource to the sector, and might alleviate confusion around how the two reviews are related (if they were in separate papers), but would it be worth adding a sentence in the introduction as to why these have been combined and not published separately? You touch on that a bit in lines 102-3 but maybe a direct statement about why they have been combined would be helpful.

• Line 738: what do you mean by menstrual or hygiene practices (also in line 591)? Is this referring to non-menstrual hygiene practices like handwashing?

• Given when this is likely to be published, would it be worth briefly mentioning the current work by the collective to develop a global definition of ‘menstrual health’ (maybe in the conclusion where you talk about the need for interdisciplinary efforts)? This might also be a good place to briefly mention the JMP 2012 definition of MHM. These two could be examples to build upon, including revisiting existing global definitions and filling in remaining gaps (such as ‘a set of core menstrual knowledge items,’ which would be incredibly helpful!).

• The article is very well-written and clear. I’m just having a bit of trouble understanding exactly what you mean in the last sentence of the conclusion – would it be worth specifying what you mean by “more attention” and clarifying the last half of that sentence?

• You give a clear recommendation to publish questionnaires in the paragraph starting on line 675 and I think this would make a huge difference in the sector (from interpretation of results to learning and supporting future measurement approaches). Would it be worthwhile to add this recommendation to the conclusion? Maybe briefly after you recommend clearly defining concepts?

Reviewer #2: This is an important contribution as measurement continues to be a major challenge within the field of menstrual health and hygiene. The author's methodology was rigorous and detailed. The paper makes a strong argument around the inconsistencies and weakness of current measures and methods currently being utilized.

6. PLOS authors have the option to publish the peer review history of their article (what does this mean?). If published, this will include your full peer review and any attached files.

Reviewer #1: Yes: Christie Chatterley

Reviewer #2: No

---

## [Editor Report · Acceptance letter]

20 May 2020

PONE-D-19-30808 

Measurement in the study of menstrual health and hygiene: A systematic review and audit 

Dear Dr. Hennegan:

I am pleased to inform you that your manuscript has been deemed suitable for publication in PLOS ONE. Congratulations! Your manuscript is now with our production department. 

With kind regards,

on behalf of

Dr. Nitika Pant Pai 

Academic Editor

PLOS ONE